# Students’, Teachers’, and Parents’ Knowledge About and Perceptions of Learning Strategies

**DOI:** 10.3390/bs15020160

**Published:** 2025-02-01

**Authors:** Amber E. Witherby, Addison L. Babineau, Sarah K. Tauber

**Affiliations:** 1Department of Psychological Sciences, Creighton University, Omaha, NE 68178, USA; 2Department of Psychology & Philosophy, Texas Woman’s University, Denton, TX 76204, USA; ababineau@twu.edu; 3Department of Psychology, Texas Christian University, Fort Worth, TX 76109, USA; uma.tauber@tcu.edu

**Keywords:** learning strategies, metacognitive knowledge, teachers, parents, survey

## Abstract

Previous research has demonstrated that students have imperfect knowledge about learning strategies. Moreover, very little is known about teachers’ and parents’ knowledge about learning strategies, which is important as these groups can help to model and teach students effective strategies. The goal of the present research was to add to this literature by measuring students’, teachers’, and parents’ beliefs about several learning strategies that have received empirical scrutiny, using methodology that builds upon prior work. To do so, participants were presented with a series of learning scenarios exemplifying a different learning strategy and rated each individually on effectiveness, familiarity, and their likelihood of using (or recommending) it in the future. Participants had accurate knowledge about effective strategies, rating retrieval practice and distributed study as the most effective learning strategies. There were variations within groups on their levels of familiarity with each strategy, which may have informed their ratings of effectiveness. For instance, participants rated interleaving as less effective compared to blocking but were also less familiar with the former. On a positive note, these outcomes suggest that people generally have good knowledge about learning strategies and underscore the importance of a broad dissemination of educational research.

## 1. Introduction

Metacognition is broadly defined as a person’s thoughts about cognitive phenomena ([11], [12]). Key components of metacognition include metacognitive knowledge (i.e., people’s knowledge about cognition), metacognitive monitoring (i.e., people’s assessments of cognitive performance), and metacognitive control (i.e., decisions people make about how to approach a cognitive task) (for reviews, see [7]; [31]). Most important for the present purposes, having accurate metacognitive knowledge is an important pre-cursor for students to use effective learning strategies. Although ample research has identified highly effective learning strategies, it is unclear how knowledgeable people are on this topic ([8]; [9]; [25]). Accordingly, we administered a survey to college students to evaluate their knowledge about and perceptions of several learning strategies. As well, given that people are not born with knowledge about how to learn, it is important to understand how (and from where) people develop this knowledge. For students, teachers and parents likely serve as the primary resource for learning how to learn. As such, it is important to evaluate whether teachers and parents have accurate knowledge about learning strategies. To that end, we also evaluated teachers’ and parents’ knowledge and perceptions.

### 1.1. Which Strategies Are Effective for Learning?

To contextualize people’s knowledge about learning strategies, it is necessary to start with a brief review of the literature evaluating the effectiveness of these strategies. In general, strategies that are more passive on the students’ part (e.g., rereading, highlighting) tend to be less effective compared to strategies that require active engagement with the content (e.g., retrieval practice, elaborative interrogation) ([8]; [9]). Moreover, the effectiveness of learning strategies can vary depending on characteristics of the to-be-learned material and of the learner. To attempt to quantify the effectiveness of commonly investigated learning strategies, Dunlosky and colleagues reviewed the literature for 10 learning strategies, assigning each a “utility rating” ([8]). Two strategies, retrieval practice and distributed study, received high utility ratings, indicating that their effectiveness is robust across various contexts and learners. Four strategies (blocking content order, interleaving content order, elaborative interrogation, and self-explanation) received moderate utility ratings, suggesting that they can be effective for some types of materials. Finally, six strategies (highlighting, rereading, keyword mnemonic, imagery, summarization, using a massed restudy schedule) received low utility ratings, suggesting that their effectiveness may be limited to certain contexts and/or learners.

From this review, retrieval practice and using a distributed study schedule stand above other strategies with regard to their general effectiveness. However, it becomes more challenging to draw clear conclusions about whether one strategy is more effective than another within each level of utility rating (e.g., summarization versus elaborative interrogation). Moreover, although these are the general conclusions about the effectiveness of these strategies, it is important to note that the way that students implement a strategy can moderate its effectiveness. As one example, rereading can be more effective if used in combination with a distributed restudy schedule as compared to a massed restudy schedule ([25]). Likewise, highlighting and underlining can be effective learning strategies if students are strategic with them and are able to identify information that is important ([25]). As a final example, the keyword mnemonic can be particularly helpful for learning various types of paired associates (e.g., foreign language vocabulary and first language vocabulary; for a review, see [28]; U.S. states and their capitals; [21]). Thus, when measuring people’s beliefs about learning strategies, it is important to ensure they all consider the same form of implementation when evaluating a strategy’s effectiveness. In addition, the effectiveness of a strategy may depend on students’ ability. For instance, the effectiveness of summarization is moderated by the quality of students’ summarizations ([9]). That is, students who produce high-quality summarizations will retain more compared to students who produce low-quality summarizations. Thus, although some strategies generally appear more effective than others (e.g., retrieval practice relative to rereading), we encourage readers to consider these strategies flexibly in the context of when and how they are being used.

### 1.2. Students’ Knowledge and Use of Learning Strategies

Several researchers have explored the frequency with which students report using various learning strategies ([2]; [3]; [13]; [15]; [20]; [26]; [37]; [38]). A consistent outcome from these studies is that students use a variety of learning strategies, many of which are suboptimal. For instance, as reviewed above, two of the most effective strategies for enhancing learning are distributed study (i.e., spacing study sessions out over time) and retrieval practice (i.e., self-testing). When asked how students space their study, about 52% reported cramming their studying into a single session, which is the less effective strategy compared to spacing study over multiple sessions ([15]; [20]; [26]). It is unclear why so many students report using the less effective strategy. One possibility is that students have incomplete knowledge about learning. Consistent with this possibility, of the roughly 70% of students who reported using retrieval practice, most reported using it to monitor their learning rather than to help them learn the information. Thus, students appear to have incomplete knowledge about why self-testing is beneficial. Although retrieval practice can be beneficial for helping students identify what they do and do not know, it is also beneficial in that the act of retrieving information from memory increases the memory strength of that information (e.g., [30]).

Although these studies are informative for understanding how students study, they do not provide clear insight into students’ knowledge about the effectiveness of learning strategies. That is, just because a student uses (or does not use) a strategy, it does not necessarily mean that they think that strategy is effective (or ineffective). Rather, many things can influence whether students use a particular learning strategy (e.g., the amount of time available, the amount of effort they are willing to invest) ([29]). To evaluate students’ knowledge about the effectiveness of learning strategies, [24] ([24]) presented students with a series of learning scenarios that pitted learning strategies against each other (e.g., self-testing vs. restudying; spacing vs. massing study) and had them choose the strategy that would be most beneficial for learning. Students generally did not endorse the empirically supported strategy. For instance, students rated massing as more effective than spacing and rereading as more effective than self-testing.

As reviewed above, students appear to hold inaccurate knowledge about the effectiveness of common learning strategies when they make relative comparisons; however, students’ knowledge about individual learning strategies cannot be determined in a paradigm that pits strategies against each other. For instance, students may believe that self-testing and rereading are both effective, but they may rate rereading as somewhat more effective if forced to choose between them. To evaluate students’ knowledge about learning strategies in isolation, [3] ([3]) had students rate the effectiveness of 10 common learning strategies. Students rated reading notes as the most effective strategy and self-testing as one of the least effective strategies. Relatedly, [16] ([16]) found that students underappreciated the benefits of spaced study schedules and interleaving content order (i.e., mixing problems of different types) when evaluating the utility of these strategies. Similarly, even when students are given experience with more effective (i.e., interleaved study schedule) and less effective (i.e., blocking study schedule) learning strategies, students still report a preference for the less effective strategy ([35]). Thus, students’ knowledge about the effectiveness of these learning strategies was far from perfect.

Why might students have inaccurate knowledge about learning strategies? One possibility is that they were never taught about effective learning techniques by their teachers or parents. Indeed, [26] ([26]) found that only 36% of students reported using strategies that they learned from their teacher. This low percentage is surprising given that, in surveys of college teachers, most reported discussing learning strategies in class (79%, [26]; 91%, [19]). Even so, if students learn about strategies from their teachers (or parents), it is necessary to know whether these groups have accurate knowledge about learning strategies. If not, they may not provide effective advice.

### 1.3. Teachers’ and Parents’ Knowledge About Learning Strategies

Few studies have explored teachers’ knowledge about learning strategies. The available literature suggests that teachers’ knowledge about learning strategies is also imperfect ([10]; [14]; [33]). [10] ([10]) and [14] ([14]) presented high school teachers with learning scenarios like those used by [24] ([24]) in which two learning strategies were pitted against each other, and the teachers predicted which strategy was more effective for enhancing learning. The teachers generally did not endorse the empirically supported learning strategies. For instance, only about 30% of the teachers endorsed retrieval practice and 50% of the teachers endorsed spacing study ([10]). Using similar methodology with college teachers, [26] ([26]) found somewhat more promising outcomes. Specifically, 62% of the teachers endorsed retrieval practice and 74% endorsed spacing study. In addition, the teachers reported recommending a combination of effective strategies (e.g., retrieval practice) and less effective strategies (e.g., rereading).

Parents can also serve as an important resource for students’ development of learning skills ([22]). In a survey with high school students, 35% of students on a high-achievement track (8% of students on a low-achievement track) reported seeking academic assistance from their parents ([39]). [18] ([18]) interviewed parents of children in 1st–5th grade about their involvement with their children’s homework. Most parents (97%) reported helping their children in some capacity, but many of them reported feeling unprepared. In addition, only about half of the parents (55%) reported discussing learning strategies with their children. There is evidence that parental involvement with their child’s learning is positively related to that child’s academic achievement ([6]; [23]). This may be because parents who are involved teach their children effective learning strategies.

Consistent with this possibility, [22] ([22]) found that parents’ beliefs about strategies for learning math predicted students’ use of those strategies about a year later. In their study, they surveyed parents and children (13–15 years old) at two time points about a year apart. Parents and children rated the effectiveness of 21 strategies that could be used to learn math (e.g., *I work on math problems in textbooks; I create my own math questions; when I study mathematics*, *I try to understand the relationship between different contents/topics*). As well, children also rated how frequently they used each strategy. They found that parents’ ratings of effectiveness at Time 1 predicted students’ use of the strategy at Time 2. Moreover, students’ ratings of effectiveness at Time 2 mediated this relationship, suggesting that parents’ beliefs impacted students’ beliefs, which in turn influenced their learning strategy use. Although this study highlights the role that parents’ knowledge can play in students’ knowledge of learning strategies, this study was limited to parents and children in Japan and to one content domain (math). Moreover, this study only explored whether parents’ beliefs were related to students’ beliefs and not whether parents have accurate knowledge about effective learning strategies. Critically, no research has explored the latter issue. Thus, it is unclear whether parents have accurate knowledge about learning strategies.

### 1.4. The Present Study

In sum, several questions remain about the knowledge that students, teachers, and parents hold about learning strategies. The primary motivation of the present study was to address these questions. Specifically, we aimed to address the limitations and gaps in prior work in the following ways:*Measuring participants knowledge about learning strategies without pitting strategies against each other*. By forcing participants to pick one of two strategies, it is challenging to draw conclusions about their knowledge. For instance, participants may believe both provided strategies are effective but artificially choose one over the other. In the present study, we allowed participants to rate each strategy in isolation, which allows for a clearer interpretation of their knowledge about that strategy.*Providing concrete examples about how a strategy is implemented*. Providing concrete examples ensures that participants consider each strategy similarly. As reviewed above, the way strategies are implemented can moderate their effectiveness ([8]; [25]). Thus, to accurately measure and compare participants’ ratings of a strategy, participants need to consider the same method of implementation. Moreover, providing concrete examples may be especially important if people are unfamiliar with a strategy. Thus, in the present study, we provided participants with a concrete example of one way the strategy could be implemented.*Increasing data on teachers’ knowledge about learning strategies*. As reviewed above, few studies have investigated teachers’ knowledge about learning strategies. Moreover, those that did included the limitations described in points 1 and 2[note 1]. Thus, the present study aimed to provide more data on teachers’ knowledge while addressing these issues.*Gathering novel data on parents’ knowledge about learning strategies*. No research has directly evaluated parents’ knowledge of common learning strategies. Rather, researchers have focused on parental involvement with homework and how parents’ beliefs about learning strategies in math are related to students’ use of those strategies. In the present study, we directly evaluated parents’ knowledge of the effectiveness of common learning strategies with a sample of parents with somewhat older children than has been investigated previously (i.e., middle and high school children).

We also included several additional novel features to build upon prior work. We investigated learning strategies that had not been previously investigated (e.g., interleaved restudy schedule, elaborative interrogation). In addition to rating the effectiveness of each strategy, participants rated their familiarity with it and their likelihood of using (students) or recommending (teachers and parents) it. These novel ratings are useful for providing insight into participants’ ratings of effectiveness. For instance, if a strategy is given a low effectiveness rating, it could be because participants are unaware of it (e.g., a low familiarity rating) or because they know about the strategy and believe that it is ineffective (e.g., a high familiarity rating). Additionally, students may have accurate knowledge about learning strategies, yet they still may choose not to use effective strategies because of various barriers such as time cost or low self-efficacy ([29]).

## 2. Materials and Methods

### 2.1. Participants

We used past research on surveys evaluating the effectiveness of learning strategies as a guide for our target sample size ([3]; [26]). Our target goal was to recruit 150 participants per group. Consistent with prior research, we collected our student sample from an undergraduate university research participation pool. To obtain a representative and diverse sample for our parent and teacher samples, we recruited participants from multiple sources (see below).

A total of 151 undergraduate students were recruited from the Texas Christian University (TCU) psychology research pool and participated in exchange for partial course credit. Students were college-aged (*M* = 20.23, *SE* = 0.29) and predominantly identified as a woman and White (see Table 1 for more detailed demographic information). Students had completed an average of 2 years of college (i.e., total years of education, *M* = 13.94, *SE* = 0.11) and reported an average college GPA of 3.4 (*SE* = 0.04). Most students (84.1%, *n* = 127) reported that they were not first-generation college students (i.e., most students had at least one parent with a college degree).

A total of 169 parents completed the survey and were paid $10 (USD) for their participation. Parents were recruited from a TCU community forum (*n* = 11), ads posted on Facebook (*n* = 12), Amazon’s Mechanical Turk (mTurk) Cloud Research (*n* = 118)[note 2], and a laboratory database (*n* = 28).[note 3] We had four inclusion criteria for the parent sample: (a) have a child aged 10 to 17 years old, (b) have no current or prior experience as a teacher, (c) speak English, and (d) live in the United States. Of the 169 parents, all responses from 26 (15.4%) were excluded because the participant indicated they had current or prior experience teaching (laboratory database, *n* = 13; mTurk, *n* = 9; TCU community forum, *n* = 4)[note 4]. Thus, the final parent sample consisted of 143 participants. Parents were middle-aged (*M* = 42.50, *SE* = 1.91) and predominantly identified as a woman and White (see Table 1). Most parents reported having at least some college experience (83.9%). Parents also reported demographic information about their child. The average age of the parents’ children was 13.54 years old (*SE* = 0.19), and the majority were identified as a boy (67.8%, *n* = 97). The children ranged from being in 4th–12th grade.

A total of 194 teachers were recruited and were paid $10 (USD) for their participation. Teachers were recruited via targeted e-mails[note 5] (*n* = 99), mTurk Cloud Research (*n* = 44), ads posted on Facebook (*n* = 28), from a local middle school (*n* = 15), and via referral from other teachers (*n* = 8). We had three inclusion criteria for the teacher sample: (a) they had to have experience working as a teacher at a middle school, high school, or college, (b) speak English, and (c) live in the United States. Prior to completing the survey, teachers were required to indicate the school at which they taught and the grade level(s) they taught. Of the 194 teachers, all responses from 3 (1.5%) were excluded because the participant did not include any information about their teaching experience (Facebook ad, *n* = 2; referral *n* = 1). Thus, the final teacher sample consisted of 191 participants. The average age for the teacher sample was 40.04 (*SE* = 0.82), and they predominantly identified as a woman and White (see Table 1). All teachers held an advanced degree. On average, teachers in the sample had been teaching for more than a decade (*M* = 11.97 years, *SE* = 0.63). The teachers taught at a range of levels (college or university, *n* = 87; high school, *n* = 46, middle school, *n* = 54; multiple levels, *n* = 4) and taught a variety of topics including science (*n* = 59; e.g., biology, engineering, psychology, computer science), math (*n* = 53; e.g., algebra, statistics, calculus, geometry), English (*n* = 20; e.g., language arts, English as a second language, literature), social studies or liberal arts (*n* = 20; e.g., history, anthropology, government), as well as others (*n* = 52; e.g., foreign language, arts, business)[note 6]. The Texas Christian University Institutional Review Board approved this study.

### 2.2. Materials and Procedure

The survey was nearly identical for all groups. A copy of the questions that each group received can be found in the online supplemental materials “https://osf.io/ghtyn/”. Participants were presented with a series of 12 learning scenarios that each described a hypothetical student using one of twelve learning strategies (see Table 2). The learning strategies were selected based on common strategies that students report using and that have received empirical scrutiny (for a review, see [8]).

Before beginning the survey, participants were told that we were interested in understanding their beliefs about the effectiveness of various learning strategies. Specifically, students received the following instructions:
“People have different ideas about what strategies are beneficial for student learning. Given that you (as a student) are often faced with the challenging task of learning information for class, we are interested in your beliefs about which strategies are beneficial for learning. In the following survey, we will give you examples of different learning strategies and ask you to rate them.”
Teachers received these instructions:
“People have different ideas about what strategies are beneficial for student learning. Given that you (as a teacher) often serve as a primary resource for your students when completing homework or studying for exams, we are interested in your beliefs about which strategies are beneficial for learning. In the following survey, we will give you examples of different learning strategies and ask you to rate them.”
Finally, parents received these instructions:
“People have different ideas about what strategies are beneficial for student learning. Given that you (as a parent) often serve as a primary resource for your child when completing homework or studying for exams, we are interested in your beliefs about which strategies are beneficial for learning. In the following survey, we will give you examples of different learning strategies and ask you to rate them.”

The survey was entirely self-paced. After reading about each strategy, participants were asked to rate the effectiveness of the strategy on a scale from 1 (*very ineffective*) to 10 (*very effective*) and their familiarity with the strategy on a scale from 1 (*very unfamiliar*) to 10 (*very familiar*). Students were asked how likely they would be to use the strategy, whereas parents and teachers were asked how likely they would be to recommend the strategy to their child or student on a scale of 1 (*very unlikely*) to 10 (*very likely*). For each scenario, participants made the ratings in this order (i.e., effectiveness, familiarity, and the likelihood of use or recommendation). Ratings were made by clicking on a value from 1 to 10 (for participants who completed the survey electronically) or by circling a value from 1 to 10 (for participants who completed a physical copy of the survey).

As an additional novel component of our study, given that researchers have recently investigated whether the effectiveness of retrieval practice is moderated by retrieval format, we explored whether students’ and teachers’ knowledge about retrieval practice was influenced by retrieval format (i.e., overt vs. covert; [32]). As such, students and teachers were presented with an additional two learning scenarios (see Table 3). One scenario described overt retrieval (i.e., retrieval practice by physically recalling an item), whereas the other described covert retrieval (i.e., retrieval practice by mentally recalling an item).

Most participants (i.e., all students and parents and 92.3% of teachers) completed the survey on Qualtrics. All students and 15 parents completed the survey in the lab. The remaining parents and teachers completed the survey on their own time from their own personal computers. For these participants (i.e., all of whom completed the survey on Qualtrics), the order of the learning scenarios was quasi-randomized in a new order for each participant. Specifically, the order of the 12 primary learning strategies was randomized anew for each participant with the constraint that, for students and teachers, the overt retrieval scenario and the covert retrieval scenario immediately followed the retrieval practice scenario. The order of the overt and covert scenarios was randomized anew for each participant.

Fifteen teachers (i.e., 7.7%) completed the survey using a physical paper copy, which they completed on their own time and mailed back to the researchers. The order of the learning scenarios was quasi-randomized. Specifically, participants completed the survey in the same random order, with the constraint that the overt retrieval scenario and the covert retrieval scenario immediately followed the retrieval practice scenario. After participants rated the retrieval practice scenario, they rated the overt retrieval practice scenario, followed by the covert retrieval practice scenario.[note 7]

Students and teachers were given an additional six multiple-choice questions to evaluate their beliefs about notetaking and learning styles. These questions were adapted from [34] ([34]). Specifically, students were asked how frequently (on a 5-point scale from *never* to *always*) their professors (a) use PowerPoint to present information to their class, (b) give them instructions regarding how they should take notes, and (c) give them handouts to facilitate their notetaking. Teachers received these same questions, except they were framed around how often they did each of these things (e.g., *how often do you give your students instruction regarding how they should take notes?*). Students were also asked to estimate the percentage of information from class that they record in their notes (teachers were asked to estimate the percentage of information from class that they thought students recorded in their notes). Students and teachers responded to this question by selecting one of five options: 0–20%, 21–40%, 41–60%, 61–80%, 81–100%. The order of the notetaking questions was randomized for participants who completed the survey on Qualtrics and fixed in the order described for teachers who completed the survey with a physical copy. Next, students and teachers were asked whether they believe students have different learning styles (e.g., visual vs. auditory learners) and responded by selecting “yes” or “no”. Finally, students were asked whether they think professors should teach in a way that accommodates those differences (teachers were asked whether they teach in a way that accommodates those differences) and responded by selecting, “yes”, “no”, or “I did not say yes”. The response options for these six multiple-choice questions were presented in a fixed order.[note 8]

Finally, all participants answered a series of demographic questions (e.g., age, gender, ethnicity). Teachers were asked questions about their teaching (e.g., how many years they have been teaching, what grade levels they teach, what topics they teach). In addition, students and teachers were asked to rate how confident they are that they know which strategies are effective for learning and responded by selecting one of five options: 0–20%, 21–40%, 41–60%, 61–80%, 81–100%. Students and teachers were also asked whether they have previously received special training on student learning strategies and responded by selecting “yes” or “no”. If they selected yes, they were then given a free-response question to explain. Students and teachers were asked whether they would attend a workshop to receive training on student learning strategies and responded by selecting “yes” or “no”. Teachers were asked whether they think it would be important for teachers to attend a workshop in which they receive training on student learning strategies and responded on a 5-point scale from “*strongly disagree*” to “*strongly agree*”. The order of the demographic questions (and these follow-up questions for students and teachers) was fixed in the order described.

## 3. Results

As reviewed above, ample evidence supports the conclusion that retrieval practice is a highly effective strategy for all types of learners to learn a wide variety of content ([8]). As such, throughout the results, we use retrieval practice as a baseline effective strategy to which we will compare all other strategies.[note 9] To organize the results, we conducted a series of 2 (strategy) × 3 (group) mixed ANOVAs to evaluate (a) participants’ beliefs for all strategies compared to retrieval practice and (b) whether these beliefs differed based on the sample (students, parents, teachers). Given that some strategies have a natural comparison strategy (e.g., distributed restudy schedule vs. massed restudy schedule), we also conducted 2 × 3 ANOVAs for these comparisons. Finally, given that only students and teachers answered questions about covert and overt retrieval practice, we conducted a 2 × 2 ANOVA for these comparisons. For all analyses, to follow up on significant main effects and interactions, we used the Bonferroni correction to account for multiple comparisons. As well, we checked for the assumption of normality using Levene’s test for homogeneity of variance. When this assumption was violated, we reported follow-up comparisons using a Welch’s ANOVA, which tests if the effect remains significant while accounting for unequal variances.

For interested readers, we have also conducted additional analyses that can be located in the online supplemental materials (https://osf.io/ghtyn/). Specifically, we conducted a series of one-way ANOVAs (12 levels) for each strategy separately for each measure (effectiveness, familiarity, and future use/recommendation) for each group (students, parents, and teachers). These outcomes can be found in Appendix B in the online supplemental materials. Additionally, we have also conducted a series of one-way ANOVAs (three levels) comparing each group separately for each strategy and measure. These outcomes can be found in Appendix C in the online supplemental materials.

### 3.1. Ratings of Effectiveness

In general, participants appeared to have accurate knowledge about two of the most effective strategies for improving student learning: using a distributed restudy schedule and retrieval practice (see Table 4). Across groups, participants consistently gave the highest effectiveness ratings to these strategies. Participants in all groups also consistently gave lower ratings to interleaving content order and using a massed restudy schedule. The remaining strategies all generally fell between these ends, with variability between groups in how effective participants believed they were.

#### 3.1.1. Retrieval Practice vs. Highlighting

Significant main effects of strategy, *F*(1,482) = 118.60, *p* < 0.001, η_p_^2^ = 0.20, and group, *F*(2,482) = 14.08, *p* < 0.001, η_p_^2^ = 0.06, were qualified by a significant interaction between strategy and group, *F*(2,482) = 11.49, *p* < 0.001, η_p_^2^ = 0.05. Specifically, although all groups provided higher ratings of effectiveness for retrieval practice relative to highlighting, the difference was largest for students (*d* = 0.86), followed by teachers (*d* = 0.64) and parents (*d* = 0.30).

#### 3.1.2. Retrieval Practice vs. Rereading

All groups provided higher ratings for retrieval practice (*M* = 8.51, *SE* = 0.08) compared to rereading (*M* = 7.22, *SE* = 0.10), *F*(1,482) = 133.89, *p* < 0.001, η_p_^2^ = 0.22. There was also a significant main effect of group, *F*(2,482) = 13.90, *p* < 0.001, η_p_^2^ = 0.06, indicating that students (*M* = 8.11, *SE* = 0.12) and parents (*M* = 8.11, *SE* = 0.13) gave higher ratings compared to teachers (*M* = 7.37, *SE* = 0.11), *p*s < 0.001. Students’ and parents’ ratings did not differ, *p* = 1.00. The interaction between strategy and group was not significant, *F*(2,482) = 7.80, *p* = 0.07, η_p_^2^ = 0.01.

#### 3.1.3. Retrieval Practice vs. Keyword Mnemonic

Significant main effects of strategy, *F*(1,482) = 80.05, *p* < 0.001, η_p_^2^ = 0.14, and group, *F*(2,482) = 14.49, *p* < 0.001, η_p_^2^ = 0.04, were qualified by a significant interaction between strategy and group, *F*(2,482) = 3.32, *p* = 0.04, η_p_^2^ = 0.01. Specifically, although all groups rated retrieval practice as more effective compared to the keyword mnemonic, the difference was largest for parents (*d* = 0.65), followed by teachers (*d* = 0.44) and students (*d* = 0.39).

#### 3.1.4. Retrieval Practice vs. Imagery

A significant main effect of strategy, *F*(1,481) = 38.32, *p* < 0.001, η_p_^2^ = 0.07, and a non-significant main effect of group, *F*(2,481) = 2.47, *p* = 0.09, η_p_^2^ = 0.01, were qualified by a significant interaction between strategy and group, *F*(2,481) = 5.13, *p* = 0.006, η_p_^2^ = 0.02. Students and parents both provided higher ratings for retrieval practice compared to imagery, *p*s < 0.004. Teachers’ ratings did not differ for these strategies, *p* = 0.08[note 10].

#### 3.1.5. Retrieval Practice vs. Summarization

A significant main effect of strategy, *F*(1,482) = 15.51, *p* < 0.001, η_p_^2^ = 0.03, and a non-significant main effect of group, *F*(2,482) = 1.72, *p* = 0.18, η_p_^2^ = 0.01, were qualified by a significant interaction between strategy and group, *F*(2,482) = 7.55, *p* < 0.001, η_p_^2^ = 0.03. Both students and parents provided higher ratings for retrieval practice relative to summarization, *p*s < 0.001. Teachers’ ratings did not differ between these strategies, *p* = 0.40.

#### 3.1.6. Retrieval Practice vs. Massed Restudy Schedule

All groups gave higher ratings of effectiveness to retrieval practice (*M* = 8.51, *SE* = 0.08) relative to using a massed restudy schedule (*M* = 4.61, *SE* = 0.11), *F*(1,482) = 892.36, *p* < 0.001, η_p_^2^ = 0.65. A significant main effect of group, *F*(2,482) = 12.83, *p* < 0.001, η_p_^2^ = 0.05, indicated that both students (*M* = 6.77, *SE* = 0.12) and parents (*M* = 6.80, *SE* = 0.12) gave overall higher ratings compared to teachers (*M* = 6.11, *SE* = 0.10), *p*s < 0.001. Students’ and parents’ overall ratings did not differ, *p* = 1.00. The interaction was not significant, *F*(2,482) = 1.51, *p* = 0.22, η_p_^2^ = 0.01.

#### 3.1.7. Retrieval Practice vs. Blocked Content Order

Participants’ ratings of effectiveness were higher for retrieval practice (*M* = 8.51, *SE* = 0.08) compared to blocking content order (*M* = 7.48, *SE* = 0.10), *F*(1,482) = 80.32, *p* < 0.001, η_p_^2^ = 0.14. There was also a significant main effect of group, *F*(2,482) = 4.16, *p* = 0.02, η_p_^2^ = 0.02, which indicated that parents (*M* = 8.15, *SE* = 0.12) gave overall higher ratings compared to teachers (*M* = 7.75, *SE* = 0.10), *p* = 0.03. Students’ (*M* = 8.09, *SE* = 0.11) ratings did not differ from parents’, *p* = 1.00, or teachers’, *p* = 0.08. The interaction was not significant, *F*(2,482) = 2.63, *p* = 0.07, η_p_^2^ = 0.01.

#### 3.1.8. Retrieval Practice vs. Interleaved Content Order

Significant main effects of strategy, *F*(1,482) = 720.73, *p* < 0.001, η_p_^2^ = 0.60, and group, *F*(2,482) = 49.30, *p* < 0.001, η_p_^2^ = 0.05, were qualified by a significant interaction between strategy and group, *F*(2,482) = 20.00, *p* < 0.001, η_p_^2^ = 0.08. This interaction revealed that the difference in ratings between strategies was greatest for students (*d* = 1.56), followed by teachers (*d* = 1.20) and parents (*d* = 1.14).

#### 3.1.9. Retrieval Practice vs. Elaborative Interrogation

A significant main effect of strategy, *F*(1,481) = 56.41, *p* < 0.001, η_p_^2^ = 0.11, and a non-significant main effect of group, *F*(2,481) = 0.10, *p* = 0.91, η_p_^2^ = 0.00, were qualified by a significant interaction between strategy and group, *F*(2,481) = 12.62, *p* < 0.001, η_p_^2^ = 0.05. Both students and parents gave higher ratings to retrieval practice relative to elaborative interrogation, *p*s < 0.001. By contrast, teachers’ ratings did not differ, *p* = 0.59[note 11].

#### 3.1.10. Retrieval Practice vs. Self-Explanation

A significant main effect of strategy, *F*(1,482) = 185.18, *p* < 0.001, η_p_^2^ = 0.29, and a non-significant main effect of group, *F*(2,482) = 0.15, *p* = 0.86, η_p_^2^ = 0.001, were qualified by a significant interaction between strategy and group, *F*(2,482) = 15.22, *p* < 0.001, η_p_^2^ = 0.06. The interaction demonstrated that although all groups rated retrieval practice as more effective compared to self-explanation, this difference was largest for students (*d* = 0.98), followed by parents (*d* = 0.85) and teachers (*d* = 0.38).

#### 3.1.11. Retrieval Practice vs. Distributed Restudy Schedule

A significant main effect of group, *F*(2,482) = 3.95, *p* = 0.02, η_p_^2^ = 0.02, and a non-significant main effect of strategy, *F*(1,482) = 1.69, *p* = 0.19, η_p_^2^ = 0.003, were qualified by a significant interaction between strategy and group, *F*(2,482) = 5.03, *p* = 0.007, η_p_^2^ = 0.02. The interaction revealed that students’ and parents’ ratings of effectiveness for retrieval practice and using a distributed restudy schedule did not differ, *p*s > 0.38, whereas teachers rated using a distributed restudy schedule as more effective compared to using retrieval practice, *p* < 0.001.

#### 3.1.12. Overt Retrieval Practice. vs. Covert Retrieval Practice

Overall, participants rated overt retrieval practice (*M* = 8.56, *SE* = 0.09) as more effective compared to covert retrieval practice (*M* = 7.77, *SE* = 0.11), *F*(1, 339) = 72.80, *p* < 0.001, η_p_^2^ = 0.18. There was also a significant main effect of group, *F*(1,339) = 15.06, *p* < 0.001, η_p_^2^ = 0.04, indicating that students (*M* = 8.50, *SE* = 0.13) rated these strategies as more effective than did teachers (*M* = 7.84, *SE* = 0.11). The interaction between strategy and group was not significant, *F*(1,339) = 2.49, *p* = 0.12, η_p_^2^ = 0.007.

#### 3.1.13. Blocked Content Order vs. Interleaved Content Order

Significant main effects of strategy, *F*(1,482) = 346.14, *p* < 0.001, η_p_^2^ = 0.42, and group, *F*(2,482) = 14.94, *p* < 0.001, η_p_^2^ = 0.06, were qualified by a significant interaction between strategy and group, *F*(2,482) = 9.26, *p* < 0.001, η_p_^2^ = 0.04. Although all groups gave higher ratings of effectiveness for blocking content order relative to interleaving, this difference was largest for students (*d* = 1.23), followed by teachers (*d* = 0.93) and parents (*d* = 0.82).

#### 3.1.14. Distributed Restudy Schedule vs. Massed Restudy Schedule

Significant main effects of strategy, *F*(1,482) = 876.57, *p* < 0.001, η_p_^2^ = 0.65, and group, *F*(2,482) = 5.21, *p* = 0.006, η_p_^2^ = 0.02, were qualified by a significant interaction between strategy and group, *F*(2,482) = 26.25, *p* = 0.003, η_p_^2^ = 0.02. Although all participants reported that using a distributed restudy schedule was more effective compared to using a massed restudy schedule, this difference was largest for teachers (*d* = 1.51), followed by students (*d* = 1.45) and parents (*d* = 1.26).

### 3.2. Ratings of Familiarity

Across groups, familiarity ratings were generally highest for retrieval practice and highlighting (see Table 5). By contrast, participants reported being less familiar with strategies such as interleaving and self-explanation, both of which can be effective for learning in some contexts. Lower familiarity with these strategies may partially explain why participants tended to give lower estimates of effectiveness for them compared to the other strategies.

#### 3.2.1. Retrieval Practice vs. Highlighting

Non-significant main effects of strategy, *F*(1,482) = 0.14, *p* = 0.71, η_p_^2^ = 0.00, and group, *F*(2,482) = 0.19, *p* = 0.82, η_p_^2^ = 0.001, were qualified by a significant interaction between strategy and group, *F*(2,482) = 3.95, *p* = 0.02, η_p_^2^ = 0.02. Students were more familiar with retrieval practice compared to highlighting, *p* = 0.02, whereas parents’ and teachers’ ratings of familiarity for these strategies did not differ, *p*s > 0.09.

#### 3.2.2. Retrieval Practice vs. Rereading

A significant main effect of strategy, *F*(1,482) = 30.98, *p* < 0.001, η_p_^2^ = 0.06, and a non-significant main effect of group, *F*(2,482) = 0.27, *p* = 0.77, η_p_^2^ = 0.001, were qualified by a significant interaction between strategy and group, *F*(2,482) = 3.30, *p* = 0.04, η_p_^2^ = 0.01. Both students and teachers reported being more familiar with retrieval practice relative to rereading, *p*s < 0.001. By contrast, parents’ ratings of familiarity did not differ, *p* = 0.24.

#### 3.2.3. Retrieval Practice vs. Keyword Mnemonic

Significant main effects of strategy, *F*(1,482) = 122.41, *p* < 0.001, η_p_^2^ = 0.20, and group, *F*(2,482) = 8.81, *p* < 0.001, η_p_^2^ = 0.04, were qualified by a significant interaction between strategy and group, *F*(2,482) = 4.25, *p* = 0.02, η_p_^2^ = 0.02. Although all groups reported being more familiar with retrieval practice relative to the keyword mnemonic, this difference was largest for teachers (*d* = 0.69), followed by parents (*d* = 0.63) and students (*d* = 0.43).

#### 3.2.4. Retrieval Practice vs. Imagery

A significant main effect of strategy, *F*(1,481) = 106.46, *p* < 0.001, η_p_^2^ = 0.18, and a non-significant main effect of group, *F*(2,481) = 0.01, *p* = 1.00, η_p_^2^ = 0.00, were qualified by a significant interaction between strategy and group, *F*(2,481) = 4.06, *p* = 0.02, η_p_^2^ = 0.02. Although all groups reported being more familiar with retrieval practice relative to imagery, the difference was largest for students (*d* = 0.75), followed by teachers (*d* = 0.54) and parents, (*d* = 0.37).

#### 3.2.5. Retrieval Practice vs. Summarization

Significant main effects of strategy, *F*(1,482) = 68.34, *p* < 0.001, η_p_^2^ = 0.12, and group, *F*(2,482) = 7.34, *p* < 0.001, η_p_^2^ = 0.03, were qualified by a significant interaction between strategy and group, *F*(2,482) = 16.31, *p* < 0.001, η_p_^2^ = 0.06. Students and parents reported being more familiar with retrieval practice relative to summarization, *p*s < 0.001. Teachers’ ratings of familiarity for these strategies did not differ, *p* = 0.40

#### 3.2.6. Retrieval Practice vs. Massed Restudy Schedule

Overall, participants reported being more familiar with retrieval practice (*M* = 8.94, *SE* = 0.07) compared to using a massed restudy schedule (*M* = 7.45, *SE* = 0.11), *F*(1,482) = 162.28, *p* < 0.001, η_p_^2^ = 0.25. The main effect of group was not significant, *F*(2,482) = 2.56, *p* = 0.08, η_p_^2^ = 0.01, nor was the interaction between strategy and group, *F*(2,482) = 1.84, *p* = 0.16, η_p_^2^ = 0.01.

#### 3.2.7. Retrieval Practice vs. Blocked Content Order

A significant main effect of strategy, *F*(1,482) = 75.83, *p* < 0.001, η_p_^2^ = 0.14, and a non-significant main effect of group, *F*(2,482) = 0.53, *p* = 0.59, η_p_^2^ = 0.002, were qualified by a significant interaction between strategy and group, *F*(2,482) = 7.76, *p* < 0.001, η_p_^2^ = 0.03. Although all groups rated retrieval practice as more familiar compared to blocking content order, this difference was largest for students (*d* = 0.73), followed by teachers (*d* = 0.37) and parents (*d* = 0.27).

#### 3.2.8. Retrieval Practice vs. Interleaved Content Order

Significant main effects of strategy, *F*(1,482) = 965.76, *p* < 0.001, η_p_^2^ = 0.67, and group, *F*(2,482) = 14.82, *p* < 0.001, η_p_^2^ = 0.06, were qualified by a significant interaction between strategy and group, *F*(2,482) = 27.06, *p* < 0.001, η_p_^2^ = 0.10. Although all groups reported being more familiar with retrieval practice relative to interleaving content order, this difference was largest for students (*d* = 1.67), followed by teachers (*d* = 1.40) and parents (*d* = 1.09).

#### 3.2.9. Retrieval Practice vs. Elaborative Interrogation

Significant main effects of strategy, *F*(1,481) = 194.57, *p* < 0.001, η_p_^2^ = 0.29, and group, *F*(2,481) = 4.38, *p* = 0.01, η_p_^2^ = 0.02, were qualified by a significant interaction between strategy and group, *F*(2,481) = 10.45, *p* < 0.001, η_p_^2^ = 0.04. Although all groups reported being more familiar with retrieval practice relative to elaborative interrogation, this difference was largest for students (*d* = 1.02), followed by parents (*d* = 0.70) and teachers (*d* = 0.51).

#### 3.2.10. Retrieval Practice vs. Self-Explanation

Significant main effects of strategy, *F*(1,482) = 352.53, *p* < 0.001, η_p_^2^ = 0.42, and group, *F*(2,482) = 5.95, *p* = 0.003, η_p_^2^ = 0.02, were qualified by a significant interaction between strategy and group, *F*(2,482) = 14.43, *p* < 0.001, η_p_^2^ = 0.06. Although all groups reported being more familiar with retrieval practice relative to self-explanation, this difference was largest for students (*d* = 1.18), followed by parents (*d* = 0.91) and teachers (*d* = 0.72).

#### 3.2.11. Retrieval Practice vs. Distributed Restudy Schedule

A significant main effect of strategy was also significant, *F*(1,482) = 13.90, *p* < 0.001, η_p_^2^ = 0.03, and a non-significant main effect of group, *F*(2,482) = 0.48, *p* = 0.62, η_p_^2^ = 0.002, was qualified by a significant interaction between strategy and group, *F*(2,482) = 3.36, *p* = 0.04, η_p_^2^ = 0.01. The interaction revealed that students reported being more familiar with retrieval practice relative to using a distributed restudy schedule, *p* < 0.001, whereas parents’ and teachers’ ratings of familiarity for these strategies did not differ, *p*s > 0.24.

#### 3.2.12. Overt Retrieval Practice. vs. Covert Retrieval Practice

Participants’ ratings of familiarity for covert and overt retrieval practice did not differ, *F*(1,339) = 1.36, *p* = 0.24, η_p_^2^ = 0.004. A significant main effect of group, *F*(1,339) = 4.75, *p* = 0.03, η_p_^2^ = 0.01, indicated that students reported being more familiar with these strategies (*M* = 8.96, *SE* = 0.13) relative to teachers (*M* = 8.58, *SE* = 0.12). The interaction between strategy and group was not significant, *F*(1,339) = 0.55, *p* = 0.46, η_p_^2^ = 0.002.

#### 3.2.13. Blocked Content Order vs. Interleaved Content Order

Significant main effects of strategy, *F*(1,482) = 603.33, *p* < 0.001, η_p_^2^ = 0.56, and group, *F*(2,482) = 24.22, *p* < 0.001, η_p_^2^ = 0.09, were qualified by a significant interaction between strategy and group, *F*(2,482) = 11.81, *p* < 0.001, η_p_^2^ = 0.05. Although all groups reported being more familiar with blocking content order compared to interleaving content order, this difference was largest for students (*d* = 1.38), followed by teachers (*d* = 1.11) and parents (*d* = 0.92).

#### 3.2.14. Distributed Restudy Schedule vs. Massed Restudy Schedule

All groups reported being more familiar with using a distributed restudy schedule (*M* = 8.63, *SE* = 0.08) relative to using a massed restudy schedule (*M* = 7.45, *SE* = 0.11), *F*(1,482) = 89.55, *p* < 0.001, η_p_^2^ = 0.16. The main effect of group was not significant, *F*(2,482) = 2.17, *p* = 0.12, η_p_^2^ = 0.01, nor was the interaction, *F*(2,482) = 1.26, *p* = 0.28, η_p_^2^ = 0.01.

### 3.3. Ratings of Future Use or Recommendation

Across all groups, participants gave the highest ratings for using a distributed restudy schedule and retrieval practice (see Table 6). Further, participants in all groups gave the lowest ratings to interleaving content order and using a massed restudy schedule.

#### 3.3.1. Retrieval Practice vs. Highlighting

Significant main effects of strategy, *F*(1,482) = 56.46, *p* < 0.001, η_p_^2^ = 0.11, and group, *F*(2,482) = 3.62, *p* = 0.03, η_p_^2^ = 0.02, were qualified by a significant interaction between strategy and group, *F*(2,482) = 3.62, *p* = 0.03, η_p_^2^ = 0.02. Both students, *p* < 0.001, and teachers, *p* < 0.001, gave higher ratings for retrieval practice compared to highlighting. However, parents’ recommendation ratings did not significantly differ between retrieval practice and highlighting, *p* = 0.73.

#### 3.3.2. Retrieval Practice vs. Rereading

All groups provided higher future use and recommendation ratings for retrieval practice (*M* = 8.34, *SE* = 0.08) as compared to rereading (*M* = 7.02, *SE* = 0.11), *F*(1,482) = 99.10, *p* < 0.001, η_p_^2^ = 0.17. There was a significant effect of group, *F*(2,482) = 15.07, *p* < 0.001, η_p_^2^ = 0.06[note 12], such that teachers (*M* = 7.19, *SE* = 0.12) provided lower ratings than did students (*M* = 8.00, *SE* = 0.13), *p* < 0.001, and parents (*M* = 8.01, *SE* = 0.13), *p* < 0.001. Students’ and parents’ ratings did not differ, *p* = 1.00. The interaction between strategy and group was not significant, *F*(2,482) = 2.50, *p* = 0.09, η_p_^2^ = 0.01.

#### 3.3.3. Retrieval Practice vs. Keyword Mnemonic

Significant main effects of strategy, *F*(1,482) = 84.70, *p* < 0.001, η_p_^2^ = 0.15, and group, *F*(2,482) = 9.96, *p* < 0.001, η_p_^2^ = 0.04, were qualified by a significant interaction between strategy and group, *F*(2,482) = 3.58, *p* = 0.029, η_p_^2^ = 0.02. Although all groups gave higher ratings to retrieval practice relative to the keyword mnemonic, this difference was largest for parents (*d* = 0.65), followed by teachers (*d* = 0.52) and students (*d* = 0.35).

#### 3.3.4. Retrieval Practice vs. Imagery

A significant main effect of strategy, *F*(1,482) = 50.19, *p* < 0.001, η_p_^2^ = 0.09, and a non-significant main effect of group, *F*(2,482) = 2.93, *p* = 0.06, η_p_^2^ = 0.01, were qualified by a significant interaction between strategy and group, *F*(2,482) = 8.28, *p* < 0.001, η_p_^2^ = 0.03. Students indicated higher future use ratings for retrieval practice as compared to imagery, *p* < 0.001. By contrast, both parents, *p* = 0.10, and teachers, *p* = 0.40, provided similar recommendation ratings for retrieval practice and imagery.

#### 3.3.5. Retrieval Practice vs. Summarization

A significant main effect of strategy, *F*(1,482) = 49.01, *p* < 0.001, η_p_^2^ = 0.09, and a non-significant main effect of group, *F*(2,482) = 2.01, *p* = 0.14, η_p_^2^ = 0.01, were qualified by a significant interaction between strategy and group, *F*(2,482) = 17.79, *p* < 0.001, η_p_^2^ = 0.07. The interaction revealed that students and parents provided higher ratings for retrieval practice as compared to summarization, *p*s < 0.005. Teachers did not rate retrieval practice significantly differently than summarization, *p* = 1.00.

#### 3.3.6. Retrieval Practice vs. Massed Restudy Schedule

Significant main effects of strategy, *F*(1,482) = 699.06, *p* < 0.001, η_p_^2^ = 0.59, and group, *F*(2,482) = 26.02, *p* < 0.001, η_p_^2^ = 0.10, were qualified by a significant interaction between strategy and group, *F*(2,482) = 37.17, *p* < 0.001, η_p_^2^ = 0.03. Although all groups gave higher ratings to retrieval practice compared to using a massed restudy schedule, the interaction revealed that this difference was largest for teachers (*d* = 1.41), followed by parents (*d* = 1.33) and students (*d* = 1.13).

#### 3.3.7. Retrieval Practice vs. Blocked Content Order

Significant main effects of strategy, *F*(1,482) = 90.09, *p* < 0.001, η_p_^2^ = 0.16, and group, *F*(2,482) = 3.64, *p* = 0.03, η_p_^2^ = 0.02, were qualified by a significant interaction between strategy and group, *F*(2,482) = 5.05, *p* = 0.007, η_p_^2^ = 0.02. The interaction revealed that although all groups gave higher ratings to retrieval practice compared to blocking content order, this difference was largest for students (*d* = 0.80), followed by parents (*d* = 0.44) and teachers (*d* = 0.43).

#### 3.3.8. Retrieval Practice vs. Interleaved Content Order

Significant main effects of strategy, *F*(1,482) = 893.87, *p* < 0.001, η_p_^2^ = 0.65, and group, *F*(2,482) = 25.30, *p* < 0.001, η_p_^2^ = 0.10, were qualified by a significant interaction between strategy and group, *F*(2,482) = 28.32, *p* < 0.001, η_p_^2^ = 0.11. Although all groups gave higher ratings to retrieval practice compared to interleaving content order, this difference was largest for students (*d* = 1.69), followed by teachers (*d* = 1.30) and parents (*d* = 1.12).

#### 3.3.9. Retrieval Practice vs. Elaborative Interrogation

Significant main effects of strategy, *F*(1,482) = 84.32, *p* < 0.001, η_p_^2^ = 0.15, and group, *F*(2,482) = 4.84, *p* = 0.008, η_p_^2^ = 0.02, were qualified by a significant interaction between strategy and group, *F*(2,482) = 21.31, *p* < 0.001, η_p_^2^ = 0.08. Students and parents provided higher ratings for retrieval practice as compared to elaborative interrogation, *p*s < 0.001. Teachers did not rate retrieval practice significantly differently than elaborative interrogation, *p* = 1.00.

#### 3.3.10. Retrieval Practice vs. Self-Explanation

Significant main effects of strategy, *F*(1,482) = 208.81, *p* < 0.001, η_p_^2^ = 0.30, and group, *F*(2,482) = 4.10, *p* = 0.02, η_p_^2^ = 0.02, were qualified by a significant interaction between strategy and group, *F*(2,482) = 21.64, *p* < 0.001, η_p_^2^ = 0.08. Although all groups provided higher ratings for retrieval practice compared to self-explanation, the interaction revealed that this difference was largest for students (*d* = 1.15), followed by parents (*d* = 0.84) and teachers (*d* = 0.40).

#### 3.3.11. Retrieval Practice vs. Distributed Restudy Schedule

Non-significant main effects of strategy, *F*(1,482) = 0.82, *p* = 0.37, η_p_^2^ = 0.002, and group, *F*(2,482) = 0.62, *p* = 0.54, η_p_^2^ = 0.003, were qualified by a significant interaction between strategy and group, *F*(2,482) = 13.73, *p* < 0.001, η_p_^2^ = 0.05. The interaction revealed that students provided higher future use ratings for retrieval practice as compared to a distributed restudy schedule, *p* < 0.001. By contrast, teachers provided higher recommendation ratings for a distributed restudy schedule as compared to retrieval practice, *p* = 0.01. Parents’ ratings did not significantly differ between retrieval practice and a distributed restudy schedule, *p* = 1.00.

#### 3.3.12. Overt Retrieval Practice. vs. Covert Retrieval Practice

Participants provided higher future use and recommendation ratings for overt retrieval practice (*M* = 8.17, *SE* = 0.11) as compared to covert retrieval practice (*M* = 7.56, *SE* = 0.12), *F*(1,339) = 28.38 *p* < 0.001, η_p_^2^ = 0.08. There was also a significant effect of group, *F*(1,339) = 9.76, *p* = 0.002, η_p_^2^ = 0.03, such that students (*M* = 8.21, *SE* = 0.15) provided higher ratings than did teachers (*M* = 7.57, *SE* = 0.14). The interaction between strategy and group was not significant, *F*(1,339) = 2.79, *p* = 0.10, η_p_^2^ = 0.01.

#### 3.3.13. Blocked Content Order vs. Interleaved Content Order

Significant main effects of strategy, *F*(1,482) = 382.66, *p* < 0.001, η_p_^2^ = 0.44, and group, *F*(2,482) = 32.98, *p* < 0.001, η_p_^2^ = 0.12, were qualified by a significant interaction between strategy and group, *F*(2,482) = 10.10, *p* < 0.001, η_p_^2^ = 0.04. Although all groups gave higher ratings for a blocked content order as compared to an interleaved content order, this difference was largest for students (*d* = 1.31), followed by teachers (*d* = 1.05) and parents (*d* = 0.79).

#### 3.3.14. Distributed Restudy Schedule vs. Massed Restudy Schedule

Significant main effects of strategy, *F*(1,482) = 598.00, *p* < 0.001, η_p_^2^ = 0.55, and group, *F*(2,482) = 8.08, *p* < 0.001, η_p_^2^ = 0.03, were qualified by a significant interaction between strategy and group, *F*(2,482) = 23.28, *p* < 0.001, η_p_^2^ = 0.09. Although all groups provided higher ratings for using a distributed restudy schedule relative to using a massed restudy schedule, the interaction revealed that this difference was largest for teachers (*d* = 1.52), followed by parents (*d* = 1.32) and students (*d* = 0.91).

## 4. Discussion

We evaluated students’, teachers’, and parents’ knowledge and perceptions about several common learning strategies by providing participants with concrete examples of how each strategy could be implemented and asking them to rate each on effectiveness, familiarity, and their likelihood of using or recommending it.

Prior research suggested that students typically underestimated the impact of highly effective strategies such as using retrieval practice and distributing study sessions over time ([3]; [24]). By contrast, our outcomes demonstrated that students (and parents and teachers) hold accurate knowledge about these strategies. These strategies received the highest effectiveness ratings, and participants generally rated retrieval practice as more effective compared to all other strategies. In addition, when directly comparing retrieval practice and using a distributed restudy schedule, there were minimal differences in participants’ ratings of effectiveness. This difference between the outcomes of the present study and those of prior research may have arisen due to the differences in methodology. For instance, in the present research, participants were allowed to rate each strategy independently, rather than pitting it against another strategy that they may have also believed was effective. When rating each strategy in isolation, the students provided higher ratings for effective strategies (such as self-testing) compared to less effective strategies (such as rereading), whereas in research where these strategies were pitted against each other, the students reported rereading to be more effective than self-testing ([24]). Using a paradigm where participants compared two different strategies may have biased their ratings because of the evaluative comparison of those two strategies. Relatedly, [3] ([3]) had students rate individual learning strategies in isolation and found that the students provided lower ratings of effectiveness for practice testing compared to other less effective strategies (e.g., reading notes, copying notes, highlighting notes), which contrasts with the present outcomes. However, the study by [3] ([3]) did not provide concrete examples for strategy implementation. Given that students sometimes view retrieval practice as a monitoring tool rather than a learning strategy, this may explain why the students in this study provided lower ratings of effectiveness for it as a learning strategy ([20]). More optimistically, differences between the present study and past research may also have been driven by greater dissemination of evidenced-based learning practices through popular media since the publication of those studies (2011 and 2017). For instance, books like *Make it Stick: The Science of Successful Learning* ([4]) and *Powerful Teaching: Unleash the Science of Learning* ([1]) have made research outcomes on education more accessible. This may have had the desired goal of leading people to become more informed about effective learning strategies.

Although students appeared to have accurate knowledge about effective strategies, they also appeared to have some inaccurate knowledge, overestimating the effectiveness of some strategies that are less effective for learning and comprehension (e.g., highlighting rereading). Even though participants’ ratings for these strategies was statistically lower compared to retrieval practice, numerically, they still received relatively high ratings (all above 6.5 on a 10-point scale). This outcome could have negative implications for students’ learning. Strategies such as rereading and highlighting are less effortful compared to the more effective strategies like retrieval practice. As such, if students think these strategies are somewhat effective, they may choose to use them more often than retrieval practice. This may be especially likely because students’ perceived time costs introduce a significant barrier for their decisions to use more effective learning strategies ([29]). In addition, the students also provided very low effectiveness ratings for interleaving content order during study (significantly lower than both retrieval practice and blocking content order), which in some contexts can be an effective strategy ([16]). Familiarity ratings may give insight into this outcome. Specifically, students (and teachers and parents) appear to be relatively unfamiliar with interleaving as a learning strategy. Thus, they may think it is an ineffective strategy simply because they have never learned about it.

In addition to developing a better understanding of students’ knowledge about learning strategies, key contributions of the present research are the outcomes relating to teachers and parents. Although teachers and parents play a key role in students’ development of knowledge about learning strategies, surprisingly little research has evaluated the accuracy of their knowledge about learning strategies ([22]). The outcomes from these groups were largely consistent with those reported with students. The teachers and parents provided high ratings of effectiveness for retrieval practice and using a distributed restudy schedule, yet still provided relatively high ratings for some of the less effective strategies (e.g., highlighting and rereading). This could be problematic if teachers and parents recommend less effective strategies to their students/children. Both groups reported being most likely to recommend retrieval practice and distributed restudy schedules, which is promising. Even so, they also reported being almost as likely to recommend less effective strategies such as highlighting (parents) and summarization (teachers). A key goal for education researchers will be to find ways to broadly disseminate their research to provide access to teachers and parents so that they can update their knowledge accordingly. In turn, teachers and parents will be able to make evidence-based recommendations to their students and children.

The outcomes of the present research have important applied implications. First, the methodological choices we made allowed us to evaluate participants’ knowledge of strategies in isolation, which may more clearly mirror how students consider them in real learning contexts. For instance, when making study decisions when studying for classes, students are unlikely to have two strategies pitted against each other from which they choose which to use. Instead, students likely rely on their knowledge of a particular strategy to determine whether they will use it. This decision can be influenced by several factors (e.g., how effective they think it is, whether they have used it before, if it was recommended to them, if they have enough time to use it). Thus, the way we measured students’ knowledge may be more likely to generalize to applied contexts compared to methodologies that have pitted strategies against each other.

A second applied implication relates to participants’ familiarity with the strategies. The present data suggest that, for each group of participants, there is variability in people’s familiarity with these strategies. Thus, simply telling students (or teachers and parents) that a strategy (e.g., self-explanation) is effective may not be enough if they are unfamiliar with the strategy and do not understand how to implement it appropriately. Instead, the present outcomes suggest that when educating people about learning strategies (e.g., teachers telling students which strategies are effective; professional development workshop leaders educating teachers on effective strategies), they need to be instructed how to effectively implement them.

An important consideration for interpreting the outcomes of the present research is to note that research evaluating the effectiveness of these strategies is ongoing. Whereas the effects of some of these strategies appear to be quite robust (e.g., retrieval practice is highly effective in many contexts), for other strategies, the effectiveness has been debated ([8]). For example, in many instances, interleaving content during learning is beneficial, but, for some types of materials, blocked content during learning is ideal ([5]). Additionally, the effectiveness of the strategy may depend on the learner’s goals. For instance, interleaving may be more effective when the goal is to identify differences between items, whereas blocking may be more effective when the goal is to identify similarities amongst items ([36]). Moreover, the effectiveness of some strategies (e.g., self-explanation; highlighting) can depend on the way those strategies are used and if they are used in combination with other strategies ([25]; [27]). Our goal was not to set definitive guidelines for the effectiveness of each strategy but rather to evaluate students’, teachers’, and parents’ knowledge and perceptions of them. Thus, we encourage readers to think flexibly about the effectiveness of these strategies as empirical work continues.

Finally, it is also worth noting that each concrete example we provided participants with for each strategy was only one example of how a strategy could be implemented. Many of the strategies we investigated could be implanted in several different ways, and the ways that those strategies are implemented could moderate their effectiveness. For instance, our scenario on interleaving focused on interleaving content from two different learning domains. Implementing this strategy in this way can be less effective for learning compared to interleaving content order within a given domain (e.g., for learning differences between different related concepts; [17]). As another example, although [8] ([8]) gave the keyword mnemonic a low utility rating, it can be effective when used for certain types of materials (e.g., foreign language vocabulary; [28]). Given that our concrete example for the keyword mnemonic involved learning foreign language vocabulary (see Table 2), this could explain why participants in our study tended to rate it relatively high in effectiveness. Had we chosen a different example (e.g., learning a text passage), participants may have lowered their estimates of effectiveness. Accordingly, interpretation of participants’ ratings in the present study should be contextualized to the specific example of implementation provided rather than to other ways that strategy could be implemented. We encourage additional research to continue exploring students’, parents’, and teachers’ metacognitive knowledge related to study strategies, specifically on how the way that a strategy is implemented might impact their beliefs about the effectiveness of that strategy.

As another limitation, some of the specific wording used in our examples may have been less clear than we originally intended. This may have led to idiosyncratic differences in participants’ interpretations of our example scenarios. For instance, for the rereading scenario, we provided two possible ways to implement the strategy. One mentioned revisiting a map to study to location of the United States, whereas the other mentioned rereading a textbook to learn about each state. Given that there are multiple ways to consider the word revisiting (e.g., rereading, self-testing, distributing study), it is possible that there was ambiguity in how participants interpreted this scenario, and some participants may have considered multiple strategies when making their ratings. Thus, the wording of some scenarios, and the specific example topics chosen, pose limitations that we encourage readers to consider when contextualizing our results. To address this limitation, future research should (a) carefully evaluate the wording chosen to ensure that there is no ambiguity and (b) counterbalance the example topics chosen across strategies to ensure that participants’ ratings are representative of the strategy and not specific to the to-be-learned material in the example.

## 5. Conclusions

In sum, the present research revealed several novel outcomes that contribute to the literature regarding students’, teachers’, and parents’ knowledge about learning strategies. The accuracy of people’s knowledge appears to be mixed with some evidence of accurate knowledge (e.g., for strategies high in effectiveness) and some evidence of inaccurate knowledge (e.g., for strategies low in effectiveness). Studies that investigate people’s metacognitive knowledge are critical because students’, teachers’, and parents’ knowledge can influence the decisions that students make about how to study, which can impact their learning.

## Figures and Tables

**Table 1 behavsci-15-00160-t001:** Demographic Information.

Question	Response Options	Students	Parents	Teachers
What is your gender?	Male	21.2	32.2	36.1
	Female	78.8	67.1	61.8
	Prefer not to respond	0.0	0.7	2.1
What is your ethnicity?	Caucasian/White (non-Hispanic)	68.2	77.6	74.3
	Black (non-Hispanic)	4.6	5.6	2.6
	Asian or Pacific Islander	9.3	4.9	5.8
	American Indian	0.7	0.7	0.0
	Hispanic	15.9	7.7	11.5
	Other	1.3	3.5	2.1
	Prefer not to respond	0.0	0.0	3.7
What is your highest level of education? ^1^	Some High School	-	2.1	0.0
	Completed High School	-	11.2	0.0
	Obtained GED	-	2.8	0.0
	Some College	-	25.9	0.0
	Associate’s Degree	-	14.7	0.0
	Bachelor’s Degree	-	30.7	20.9
	Master’s Degree	-	11.9	44.5
	Doctoral Degree	-	0.7	34.0
	Prefer not to respond	-	0.0	0.5

*Note*: Value represents the percent of the sample that selected each option. ^1^ This question was only asked to parents and teachers.

**Table 2 behavsci-15-00160-t002:** Learning Scenarios used in the Present Study.

Learning Strategy	Learning Scenario
Elaborative Interrogation	One way to study is to come up with an explanation for something you are trying to learn. You can do this by asking yourself “why” questions. For example, when a student is trying to learn the second amendment of the constitution (i.e., right to bear arms), he may ask himself why this amendment was originally implemented.
Self-explanation	One way to study is to explain how you are thinking to yourself. For example, when a student is trying to learn about soundwaves, she may ask herself, “What do soundwaves mean to me?” or “What do I already know about soundwaves?”.
Summarization	One way to study is to summarize the material you are learning as you go. For example, when a student is trying to learn information from his textbook (e.g., the events leading up to World War II), he may read a section of the text, and then write a 2–3 sentence summary of the main points of that section.
Highlighting	One way to study is to highlight (or underline) information that is perceived as being important in the text or in one’s notes. For example, when a student is trying to learn about important historical figures, she may highlight (or underline) this information in her textbook while reading.
Keyword Mnemonic	One way to study is to generate keywords that link together concepts you are trying to learn. For example, perhaps a student is trying to learn Spanish vocabulary words, including the word vaca (which means cow). He may think of the word vacation in English because it sounds like vaca. Then, he thinks of an image of the two words interacting. For example, the student may think of a cow that is on vacation at a beach.
Imagery	One way to study is to create mental images as you read. For example, when a student is reading her textbook and trying to learn about the planets in the solar system, she may try to visualize all of the planets including their order relative to the sun, their color, and their size.
Rereading	One way to study is by rereading or restudying material. For example, when a student is trying to learn geography of the United States, he may study a map of the United States with all of the states labeled, and then revisit this map multiple times to study it. As another example, he may try to learn information about when each state was founded by rereading his textbook over and over.
Retrieval Practice	One way to study is by retrieving information from memory, without the answer present (i.e., practice testing). For example, when a student is trying to learn the phases of mitosis (a type of cell division), she may study by making flashcards for each phase, and then testing herself using the flashcards.
Distributed Restudy Schedule	One way to study is by spreading your study sessions out over time. For instance, if a student is learning new vocabulary words, he may choose to study them for 1 h per day, for 5 consecutive days.
Massed Restudy Schedule	One way to study is to study everything in one block of time. For instance, when learning about the different branches of government, a student may choose to study all three branches (legislative, executive, judicial) in one session for 5 h.
Interleaved Content Order	Imagine that a student has two tests next week: one in science and one in history. To study for these tests, the student may make a set of flashcards for his science test and a second set of flashcards for his history test. One way to study for these tests would be to intermix the flashcards from the two classes. That is, the student would alternate between studying flashcards for his science class and flashcards for his history class.
Blocked Content Order	Imagine that a student is trying to learn how to play basketball, which involves learning a number of different skills. One way to learn these skills is by practicing one skill at a time until it is mastered before moving on to the next skill. For example, a student may start by learning how to make a bounce pass. To do so, she would make bounce passes over and over until she has mastered that skill. Then, she may move on to learning the next skill, how to make a chest pass. She would then make chest passes over and over until she masters that skill. She would move on to the next skill (e.g., shooting free throws).

**Table 3 behavsci-15-00160-t003:** Overt Retrieval and Covert Retrieval Scenarios used in the Present Study.

Learning Strategy	Learning Scenario
Overt Retrieval Practice	When studying, a student may decide to test himself using flashcards. For example, he may put a key term on one side of the flash card and a definition on the other. One way he could test himself is by **overtly** recalling the definition for each key term. That is, when he sees a new key term, he could **say the definition out loud** before checking the definition on the other side of the flash card.
Covert Retrieval Practice	When studying, a student may decide to test herself using flashcards. For example, she may put a key term on one side of the flash card and a definition on the other. One way she could test herself is by **covertly** recalling the definition for each key term. That is, when she sees a new key term, she could recall the definition **silently in her head** before checking the definition on the other side of the flash card.

*Note*. Bold and underlined formatting reflects how these scenarios were presented to participants.

**Table 4 behavsci-15-00160-t004:** Ratings of Effectiveness.

Learning Strategy	Students (*n* = 151)	Parents (*n* = 143)	Teachers (*n* = 191)
Highlighting	6.80 (0.21)	8.06 (0.16)	6.72 (0.18)
Rereading	7.43 (0.17)	7.64 (0.17)	6.59 (0.18)
Keyword Mnemonic	8.09 (0.16)	7.20 (0.20)	7.28 (0.15)
Imagery	7.74 (0.15)	8.05 (0.14)	7.88 (0.13)
Summarization	8.14 (0.14)	7.94 (0.16)	8.28 (0.11)
Massed Restudy Schedule	4.75 (0.19)	5.01 (0.21)	4.06 (0.17)
Blocked Content Order	7.38 (0.18)	7.71 (0.17)	7.35 (0.15)
Interleaved Content Order	3.84 (0.19)	5.64 (0.21)	4.95 (0.18)
Elaborative Interrogation	7.45 (0.18)	7.52 (0.16)	8.06 (0.14)
Self-explanation	6.61 (0.19)	6.90 (0.17)	7.41 (0.14)
Distributed Restudy Schedule	8.79 (0.12)	8.45 (0.14)	8.63 (0.12)
Retrieval Practice	8.79 (0.12)	8.59 (0.13)	8.15 (0.13)
Overt Retrieval Practice	8.97 (0.11)	-	8.16 (0.13)
Covert Retrieval Practice	8.03 (0.15)	-	7.52 (0.15)

*Note:* Ratings were made on a scale from 1 (*Very Ineffective*) to 10 (*Very Effective*). Standard errors of the mean are in parentheses. Only students and teachers made ratings for overt and covert retrieval practice.

**Table 5 behavsci-15-00160-t005:** Ratings of Familiarity.

Learning Strategy	Students (*n* = 151)	Parents (*n* = 143)	Teachers (*n* = 191)
Highlighting	8.81 (0.14)	9.03 (0.14)	8.88 (0.12)
Rereading	8.36 (0.15)	8.57 (0.14)	8.38 (0.14)
Keyword Mnemonic	8.34 (0.18)	7.20 (0.23)	7.39 (0.18)
Imagery	7.64 (0.18)	8.01 (0.18)	7.90 (0.14)
Summarization	7.51 (0.20)	7.82 (0.20)	8.76 (0.11)
Massed Restudy Schedule	7.48 (0.20)	7.15 (0.21)	7.72 (0.16)
Blocked Content Order	7.66 (0.19)	8.22 (0.19)	8.24 (0.14)
Interleaved Content Order	3.07 (0.20)	5.50 (0.25)	4.31 (0.21)
Elaborative Interrogation	6.74 (0.20)	7.11 (0.21)	7.84 (0.17)
Self-explanation	5.97 (0.22)	6.48 (0.21)	7.33 (0.18)
Distributed Restudy Schedule	8.52 (0.14)	8.61 (0.15)	8.75 (0.13)
Retrieval Practice	9.15 (0.11)	8.77 (0.16)	8.91 (0.11)
Overt Retrieval Practice	9.05 (0.13)	-	8.60 (0.14)
Covert Retrieval Practice	8.88 (0.14)	-	8.56 (0.14)

*Note:* Ratings were made on a scale from 1 (*Very Unfamiliar*) to 10 (*Very Familiar*). Standard errors of the mean are in parentheses. Only students and teachers made ratings for overt and covert retrieval practice.

**Table 6 behavsci-15-00160-t006:** Ratings of the Likelihood of Future Use or Recommendation.

Learning Strategy	Students (*n* = 151)	Parents (*n* = 143)	Teachers (*n* = 191)
Highlighting	7.38 (0.21)	8.03 (0.18)	6.80 (0.19)
Rereading	7.39 (0.20)	7.52 (0.18)	6.36 (0.19)
Keyword Mnemonic	7.87 (0.21)	6.87 (0.24)	6.81 (0.17)
Imagery	7.13 (0.20)	7.92 (0.16)	7.62 (0.14)
Summarization	6.89 (0.21)	7.69 (0.19)	8.01 (0.14)
Massed Restudy Schedule	5.48 (0.23)	4.43 (0.23)	3.57 (0.18)
Blocked Content Order	6.75 (0.21)	7.52 (0.20)	7.06 (0.17)
Interleaved Content Order	2.64 (0.18)	5.26 (0.24)	3.96 (0.20)
Elaborative Interrogation	6.29 (0.21)	7.37 (0.20)	7.78 (0.16)
Self-explanation	5.55 (0.22)	6.46 (0.22)	7.11 (0.17)
Distributed Restudy Schedule	7.93 (0.16)	8.38 (0.14)	8.54 (0.13)
Retrieval Practice	8.62 (0.13)	8.50 (0.15)	8.02 (0.15)
Overt Retrieval Practice	8.40 (0.16)	-	7.94 (0.15)
Covert Retrieval Practice	8.01 (0.17)	-	7.19 (0.17)

*Note:* Ratings were made on a scale from 1 (*Very Unlikely*) to 10 (*Very Likely*). Standard errors of the mean are in parentheses. Only students and teachers made ratings for overt and covert retrieval practice.

## Data Availability

All materials, raw data, and group comparison analyses have been uploaded to the Open Science Framework and can be accessed at https://osf.io/ghtyn/. The study’s design and analyses were not pre-registered.

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
