# Peer review of "Students’, Teachers’, and Parents’ Knowledge About and Perceptions of Learning Strategies"

_behavsci, 2025, doi:10.3390/bs15020160_

Round 1
Reviewer 1 Report
Comments and Suggestions for Authors
Thank you for submitting your manuscript. It is well-written and well-structured. However, I have some questions regarding the research methodology and statistical analysis, as my focus is in educational quantitative research.
For Table 1, I recommend including the mean with the standard deviation. Also, providing only descriptive statistics is not sufficient; you should also include a table of inferential statistics, such as the results of your one-way ANOVA. This table should include values like F, p, SS, df, and MS.
In line 355, there is a major issue in your results section. You reported a t-value instead of an F-value. Are you sure you conducted a one-way ANOVA? It appears you may have used a t-test instead.
As you mentioned, you conducted a one-way ANOVA with 12 levels (strategies). Did you check for the normality and homogeneity of variance assumptions? It is unclear if your data meet the requirements for a one-way ANOVA.
In line 363, could you clarify what "ps" refers to? p value?
Could you please explain why one-way ANOVA is chosen for your research? Thank you!
Reviewer 2 Report
Comments and Suggestions for Authors
Abstract
The aim of the paper is too generic. What are these strategies used for? What is the context? How many should we think of? What do they aim to do? Why is it relevant to investigate the view of students, teachers and parents about their value?
Introduction
The readability of the introduction is low due to a lack of structure. Why not provide clear subsections. For example, by creating subsections on students, teachers and/or parents views.
There is quite a big difference in how students are able to use different effective strategies and to what extent they have knowledge about them (i.e., availability deficiency, production deficiency). Here it seems that these phenomena were not considered. It needs elaboration how this is related to fill this gap in the line of reasoning.
It is not always clear what specific statements originate from. Is this the author’s own opinion or are references missing where they are due? Like in lines 59-62.
In line 120 it was stated that ‘In sum, several questions remain about the knowledge that students, teachers, and parents hold about learning strategies’. It is, however, not clear what the benefit of this knowledge is. This should be stressed more, so that it is clear why this information is relevant to have. What could be the possible follow-up? What is the added value of having this information?
Method & Results
It is not fully clear what the added value was of the additional questionnaire on notetaking and learning styles. How does this fit into the purpose of the study? This is also not fully clear when reading the results. Could you provide more contextualization for this measure?
Discussion
The line of reasoning concerning the type of measure (i.e., individual appreciation vs. comparison) makes sense. However, if students rate the effectiveness of strategies differently due to way it is measured, does this really provide accurate info on how they perceive the strategies in real contexts? If the effect is due to the measuring, is this really a proper representation of students’ actual opinions? This needs to be elaborated upon.
